# Interpretable Nonlinear Dynamic Modeling of Neural Trajectories

**Yuan Zhao and Il Memming Park**
Department of Neurobiology and Behavior
Department of Applied Mathematics and Statistics
Institute for Advanced Computational Science
Stony Brook University, NY 11794
{yuan.zhao, memming.park}@stonybrook.edu

## Abstract

A central challenge in neuroscience is understanding how neural system implements computation through its dynamics. We propose a nonlinear time series model aimed at characterizing interpretable dynamics from neural trajectories. Our model assumes low-dimensional continuous dynamics in a finite volume. It incorporates a prior assumption about globally contractional dynamics to avoid overly enthusiastic extrapolation outside of the support of observed trajectories. We show that our model can recover qualitative features of the phase portrait such as attractors, slow points, and bifurcations, while also producing reliable long-term future predictions in a variety of dynamical models and in real neural data.

## 1 Introduction

Continuous dynamical systems theory lends itself as a framework for both qualitative and quantitative understanding of neural models [1, 2, 3, 4]. For example, models of neural computation are often implemented as attractor dynamics where the convergence to one of the attractors represents the result of computation. Despite the wide adoption of dynamical systems theory in theoretical neuroscience, solving the inverse problem, that is, reconstructing meaningful dynamics from neural time series, has been challenging. Popular neural trajectory inference algorithms often assume linear dynamical systems [5, 6] which lack nonlinear features ubiquitous in neural computation, and typical approaches of using nonlinear autoregressive models [7, 8] sometimes produce wild extrapolations which are not suitable for scientific study aimed at confidently recovering features of the dynamics that reflects the nature of the underlying computation.

In this paper, we aim to build an interpretable dynamics model to reverse-engineer the neural implementation of computation. We assume slow continuous dynamics such that the sampled nonlinear trajectory is locally linear, thus, allowing us to propose a flexible nonlinear time series model that directly learns the velocity field. Our particular parameterization yields to better interpretations: identifying fixed points and ghost points are easy, and so is the linearization of the dynamics around those points for stability and manifold analyses. We further parameterize the velocity field using a finite number of basis functions, in addition to a global contractional component. These features encourage the model to focus on interpolating dynamics within the support of the training trajectories.

## 2 Model

Consider a general $d$-dimensional continuous nonlinear dynamical system driven by external input,

$$\dot{\mathbf{x}} = F(\mathbf{x}, \mathbf{u}) \qquad (1)$$

where $\mathbf{x} \in \mathbb{R}^d$ represent the dynamic trajectory, and $F : \mathbb{R}^d \times \mathbb{R}^{d_i} \to \mathbb{R}^d$ fully defines the dynamics in the presence of input drive $\mathbf{u} \in \mathbb{R}^{d_i}$. We aim to learn the essential part of the dynamics $F$ from a collection of trajectories sampled at frequency $1/\Delta$.

Our work builds on extensive literature in nonlinear time series modeling. Assuming a separable, linear input interaction, $F(\mathbf{x}, \mathbf{u}) = F_x(\mathbf{x}) + F_u(\mathbf{x})\mathbf{u}$, a natural nonlinear extension of an autoregressive model is to use a locally linear expansion of (1) [7, 9]:

$$\mathbf{x}_{t+1} = \mathbf{x}_t + \mathbf{A}(\mathbf{x}_t)\mathbf{x}_t + \mathbf{b}(\mathbf{x}_t) + \mathbf{B}(\mathbf{x}_t)\mathbf{u}_t + \epsilon_t \tag{2}$$

where $\mathbf{b}(\mathbf{x}) = F_x(\mathbf{x})\Delta$, $\mathbf{A}(\mathbf{x}) : \mathbb{R}^d \to \mathbb{R}^{d \times d}$ is the Jacobian matrix of $F_x$ at $\mathbf{x}$ scaled by time step $\Delta$, $\mathbf{B}(\mathbf{x}) : \mathbb{R}^d \to \mathbb{R}^{d \times d_i}$ is the linearization of $F_u$ around $\mathbf{x}$, and $\epsilon_t$ denotes model mismatch noise of order $\mathcal{O}(\Delta^2)$. For example, $\{\mathbf{A}, \mathbf{B}\}$ are parametrized with a radial basis function (RBF) network in the multivariate RBF-ARX model of [10, 7], and $\{\mathbf{A}, \mathbf{b}, \mathbf{B}\}$ are parametrized with sigmoid neural networks in [9]. Note that $\mathbf{A}(\cdot)$ is not guaranteed to be the Jacobian of the dynamical system (1) since $\mathbf{A}$ and $\mathbf{b}$ also change with $\mathbf{x}$. In fact, the functional form for $\mathbf{A}(\cdot)$ is not unique, and a powerful function approximator for $\mathbf{b}(\cdot)$ makes $\mathbf{A}(\cdot)$ redundant and over parameterizes the dynamics.

Note that (2) is a subclass of a general nonlinear model:

$$\mathbf{x}_{t+1} = \mathbf{f}(\mathbf{x}_t) + \mathbf{B}(\mathbf{x}_t)\mathbf{u}_t + \epsilon_t, \tag{3}$$

where $\mathbf{f}, \mathbf{B}$ are the discrete time solution of $F_x, F_u$. This form is widely used, and called nonlinear autoregressive with eXogenous inputs (NARX) model where $\mathbf{f}$ assumes various function forms (e.g. neural network, RBF network [11], or Volterra series [8]).

We propose to use a specific parameterization,

$$\mathbf{x}_{t+1} = \mathbf{x}_t + \mathbf{g}(\mathbf{x}_t) + \mathbf{B}(\mathbf{x}_t)\mathbf{u}_t + \epsilon_t$$
$$\mathbf{g}(\mathbf{x}_t) = \mathbf{W}_g \boldsymbol{\phi}(\mathbf{x}_t) - e^{-\tau^2}\mathbf{x}_t \tag{4}$$
$$\text{vec}(\mathbf{B}(\mathbf{x}_t)) = \mathbf{W}_B \boldsymbol{\phi}(\mathbf{x}_t)$$

where $\boldsymbol{\phi}(\cdot)$ is a vector of $r$ continuous basis functions,

$$\boldsymbol{\phi}(\cdot) = (\phi_1(\cdot), \dots, \phi_r(\cdot))^\top. \tag{5}$$

Note the inclusion of a global leak towards the origin whose rate is controlled by $\tau^2$. The further away from the origin (and as $\tau \to 0$), the larger the effect of the global contraction. This encodes our prior knowledge that the neural dynamics are limited to a finite volume of phase space, and prevents solutions with nonsensical runaway trajectories.

The function $\mathbf{g}(\mathbf{x})$ directly represents the velocity field of an underlying smooth dynamics (1), unlike $\mathbf{f}(\mathbf{x})$ in (3) which can have convoluted jumps. We can even run the dynamics backwards in time, since the time evolution for small $\Delta$ is reversible (by taking $\mathbf{g}(\mathbf{x}_t) \approx \mathbf{g}(\mathbf{x}_{t+1})$), which is not possible for (3), since $\mathbf{f}(\mathbf{x})$ is not necessarily an invertible function.

Fixed points $\mathbf{x}^*$ satisfy $\mathbf{g}(\mathbf{x}^*) + \mathbf{B}(\mathbf{x}^*)\mathbf{u} = 0$ for a constant input $\mathbf{u}$. Far away from the fixed points, dynamics are locally just a flow (rectification theorem) and largely uninteresting. The Jacobian in the absence of input, $J = \frac{\partial \mathbf{g}(\mathbf{x})}{\partial \mathbf{x}}$ provides linearization of the dynamics around the fixed points (via the Hartman-Grobman theorem), and the corresponding fixed point is stable if all eigenvalues of $J$ are negative.

We can further identify fixed points, and ghost points (resulting from disappearance of fixed points due to bifurcation) from local minima of $\|\mathbf{g}\|$ with small magnitude. The flow around the ghost points can be extremely slow [4], and can exhibit signatures of computation through meta-stable dynamics [12]. Continuous attractors (such as limit cycles) are also important features of neural dynamics which exhibit spontaneous oscillatory modes. We can easily identify attractors by simulating the model.

## 3 Estimation

We define the mean squared error as the loss function

$$L(\mathbf{W}_g, \mathbf{W}_B, \mathbf{c}_{1\dots r}, \sigma_{1\dots r}) = \frac{1}{T}\sum_{t=0}^{T-1} \|\mathbf{g}(\mathbf{x}_t) + \mathbf{B}(\mathbf{x}_t)\mathbf{u}_t + \mathbf{x}_t - \mathbf{x}_{t+1}\|_2^2, \tag{6}$$

where we use normalized squared exponential radial basis functions

$$\phi_i(\mathbf{z}) = \frac{\exp\left(-\frac{\|\mathbf{z}-\mathbf{c}_i\|_2^2}{2\sigma_i^2}\right)}{\epsilon + \sum_{i=1}^{r} \exp\left(-\frac{\|\mathbf{z}-\mathbf{c}_i\|_2^2}{2\sigma_i^2}\right)}, \tag{7}$$

with centers $\mathbf{c}_i$ and corresponding kernel width $\sigma_i$. The small constant $\epsilon = 10^{-7}$ is to avoid numerical 0 in the denominator.

We estimate the parameters $\{\mathbf{W}_g, \mathbf{W}_B, \tau, \mathbf{c}, \boldsymbol{\sigma}\}$ by minimizing the loss function through gradient descent (Adam [13]) implemented within TensorFlow [14]. We initialize the matrices $\mathbf{W}_g$ and $\mathbf{W}_B$ by truncated standard normal distribution, the centers $\{\mathbf{c}_i\}$ by the centroids of the K-means clustering on the training set, and the kernel width $\sigma$ by the average euclidean distance between the centers.

# 4 Inferring Theoretical Models of Neural Computation

We apply the proposed method to a variety of low-dimensional neural models in theoretical neuroscience. Each theoretical model is chosen to represent a different mode of computation.

## 4.1 Fixed point attractor and bifurcation for binary decision-making

Perceptual decision-making and working memory tasks are widely used behavioral tasks where the tasks typically involve a low-dimensional decision variable, and subjects are close to optimal in their performance. To understand how the brain implements such neural computation, many competing theories have been proposed [15, 16, 17, 18, 19, 20, 21]. We implemented the two dimensional dynamical system from [20] where the final decision is represented by two stable fixed points corresponding to each choice. The stimulus strength (coherence) nonlinearly interacts with the dynamics (see appendix for details), and biases the choice by increasing the basin of attraction (Fig. 1). We encode the stimulus strength as a single variable held constant throughout each trajectory as in [20].

The model with 10 basis functions learned the dynamics from 90 training trajectories (30 per coherence $c = 0, 0.5, -0.5$). We visualize the log-speed as colored contours, and the direction component of the velocity field as arrows in Fig. 1. The fixed/ghost points are shown as red dots, which ideally should be at the crossing of the model nullclines given by solid lines. For each coherence, two novel starting points were simulated from the true model and the estimated model in Fig. 1. Although the model was trained with only low or moderate coherence levels where there are 2 stable and 1 unstable fixed points, it predicts bifurcation at higher coherence and it identifies the ghost point (lower right panel).

We compare the model (4) to the following "locally linear" (LL) model,

$$\begin{aligned} \mathbf{x}_{t+1} &= \mathbf{A}(\mathbf{x}_t)\mathbf{x}_t + \mathbf{B}(\mathbf{x}_t)\mathbf{u}_t + \mathbf{x}_t \\ \mathrm{vec}(\mathbf{A}(\mathbf{x}_t)) &= \mathbf{W}_A\boldsymbol{\phi}(\mathbf{x}_t) \\ \mathrm{vec}(\mathbf{B}(\mathbf{x}_t)) &= \mathbf{W}_B\boldsymbol{\phi}(\mathbf{x}_t) \end{aligned} \tag{8}$$

in terms of training and prediction errors in Table 1. Note that there is no contractional term. We train both models on the same trajectories described above. Then we simulate 30 trajectories from the true system and trained models for coherence $c = 1$ with the same random initial states within the unit square and calculate the mean squared error between the true trajectories and model-simulated ones as prediction error. The other parameters are set to the same value as training. The LL model

Table 1: Model errors

| Model | Training error | Prediction error: mean (std) |
|:---:|:---:|:---:|
| (4) | 4.06E-08 | 0.002 (0.008) |
| (8) | 2.04E-08 | 0.244 (0.816) |

has poor prediction on the test set. This is due to unbounded flow out of the phase space where the training data lies (see Fig. 6 in the supplement).

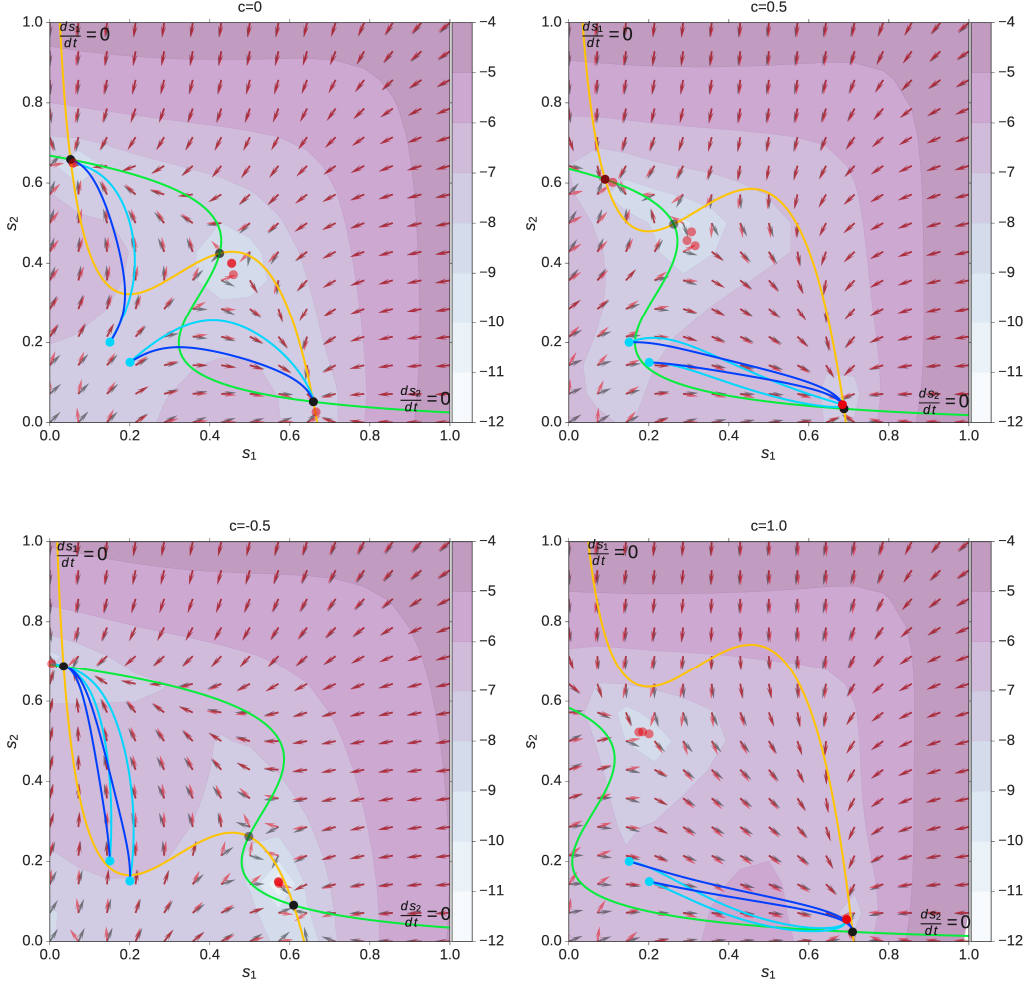

Figure 1: Wong and Wang's 2D dynamics model for perceptual decision-making [20]. We train the model with 90 trajectories (uniformly random initial points within the unit square, 0.5 s duration, 1 ms time step) with different input coherence levels $c = \{0, 0.5, -0.5\}$ (30 trajectories per coherence). The yellow and green lines are the true nullclines. The black arrows represent the true velocity fields (direction only) and the red arrows are model-predicted ones. The black and gray circles are the true stable and unstable fixed points, while the red ones are local minima of model-prediction (includes fixed points and slow points). The background contours are model-predicted $\log\|\frac{d\,\boldsymbol{s}}{d\,t}\|_2$. We simulated two 1 s trajectories each for true and learned model dynamics. The trajectories start from the cyan circles. The blue lines are from the true model and the cyan ones are simulated from trained models. Note that we do not train our model on trajectories from the bottom right condition ($c = 1$).

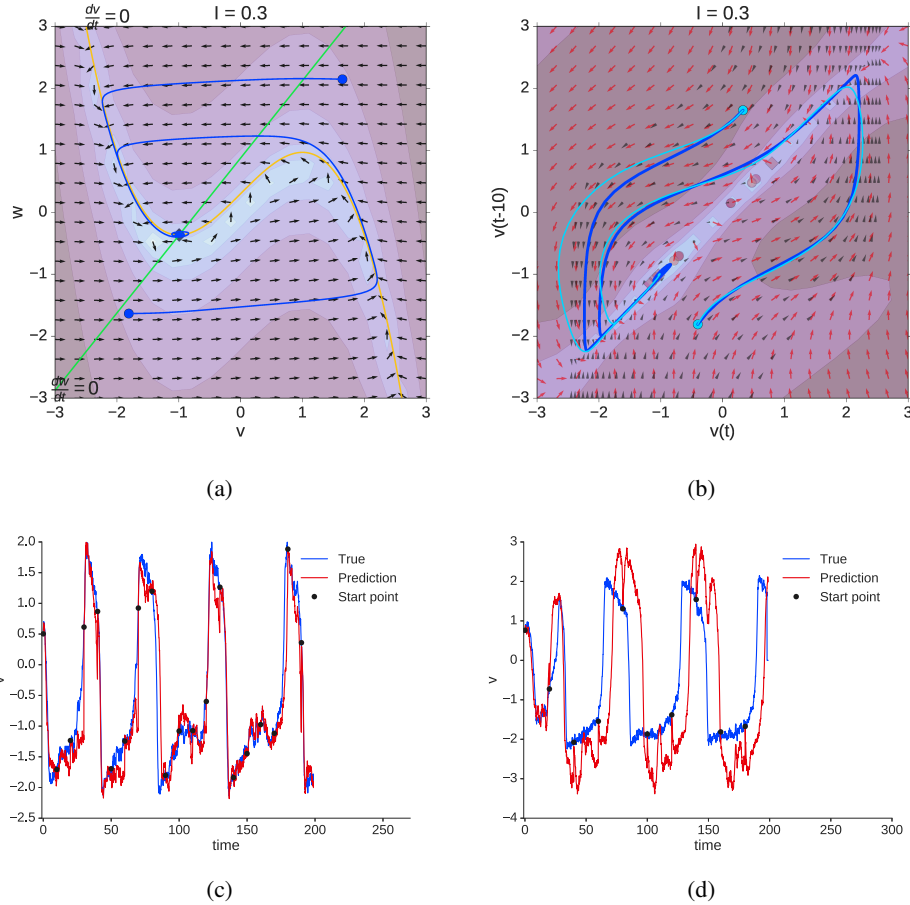

Figure 2: FitzHugh-Nagumo model. **(a)** Direction (black arrow) and log-speed (contour) of true velocity field. Two blue trajectories starting at the blue circles are simulated from the true system. The yellow and green lines are nullclines of $v$ and $w$. The diamond is a spiral point. **(b)** 2-dimensional embedding of $v$ model-predicted velocity field (red arrow and background contour). The black arrows are true velocity field. There are a few model-predicted slow points in light red. The blue lines are the same trajectories as the ones in (a). The cyan ones are simulated from trained model withe the same initial states of the blue ones. **(c)** 100-step prediction every 100 steps using a test trajectory generated with the same setting as training. **(d)** 200-step prediction every 200 steps using a test trajectory driven by sinusoid input with 0.5 standard deviation white Gaussian noise.

## 4.2 Nonlinear oscillator model

One of the most successful application of dynamical systems in neuroscience is in the biophysical model of a single neuron. We study the FitzHugh-Nagumo (FHN) model which is a 2-dimensional reduction of the Hodgkin-Huxley model [3]:

$$\dot{v} = v - \frac{v^3}{3} - w + I, \tag{9}$$

$$\dot{w} = 0.08(v + 0.7 - 0.8w), \tag{10}$$

where $v$ is the membrane potential, $w$ is a recovery variable and $I$ is the magnitude of stimulus current. The FHN has been used to model the up-down states observed in the neural time series of anesthetized auditory cortex [22].

We train the model with 50 basis functions on 100 simulated trajectories with uniformly random initial states within the unit square $[0, 1] \times [0, 1]$ and driven by injected current generated from a 0.3 mean and 0.2 standard deviation white Gaussian noise. The duration is 200 and the time step is 0.1.

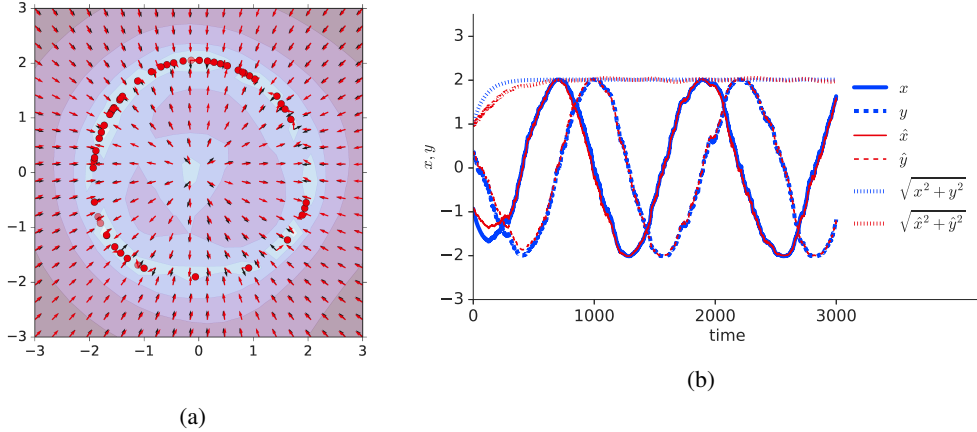

(a)　　　　　　　　　　　　(b)

Figure 3: **(a)** Velocity field (true: black arrows, model-predicted: red arrows) for both direction and log-speed; model-predicted fixed points (red circles, solid: stable, transparent: unstable). **(b)** One trajectory from the true model $(x, y)$, and one trajectory from the fitted model $(\hat{x}, \hat{y})$. The trajectory remains on the circle for both. Both are driven by the same input, and starts at same initial state.

In electrophysiological experiments, we only have access to $v(t)$, and do not observe the slow recovery variable $w$. Delay embedding allows reconstruction of the phase space under mild conditions [23]. We build a 2D model by embedding $v(t)$ as $(v(t), v(t-10))$, and fit the dynamical model (Fig. 2b). The phase space is distorted, but the overall prediction of the model is good given a fixed current (Fig. 2b). Furthermore, the temporal simulation of $v(t)$ for white noise injection shows reliable long-term prediction (Fig. 2c). We also test the model in a regime far from the training trajectories, and the dynamics does not diverge away from reasonable region of the phase space (Fig. 2d).

## 4.3 Ring attractor dynamics for head direction network

Continuous attractors such as line and ring attractors are often used as models for neural representation of continuous variables [17, 4]. For example, the head direction neurons are tuned for the angle of the animal's head direction, and a bump attractor network with ring topology is proposed as the dynamics underlying the persistently active set of neurons [24]. Here we use the following 2 variable reduction of the ring attractor system:

$$\tau_r \dot{r} = r_0 - r, \tag{11}$$

$$\tau_\theta \dot{\theta} = I(t), \tag{12}$$

where $\theta$ represents the head direction driven by input $I(t)$, and $r$ is the radial component representing the overall activity in the bump. The computational role of this ring attractor is to be insensitive to the noise in the $r$ direction, while integrating the differential input in the $\theta$ direction. In the absence of input, the head direction $\theta$ does a random walk around the ring attractor. The ring attractor consists of a continuum of stable fixed points with a center manifold.

We train the model with 50 basis functions on 150 trajectories. The duration is 5 and the time step is 0.01. The parameters are set as $r_0 = 2$, $\tau_r = 1$ and $\tau_\theta = 1$. The initial states are uniformly random within $(x, y) \in [-3, 3] \times [-3, 3]$. The inputs are constant angles evenly spaced in $[-\pi, \pi]$ with Gaussian noises ($\mu = 0, \sigma = 5$) added (see Fig. 7 in online supplement).

From the trained model, we can identify a number of fixed points arranged around the ring attractor (Fig. 3a). The true ring dynamics model has one negative eigenvalue, and one zero-eigenvalue in the Jacobian. Most of the model-predicted fixed points are stable (two negative real parts of eigenvalues) and the rest are unstable (two positive real parts of eigenvalues).

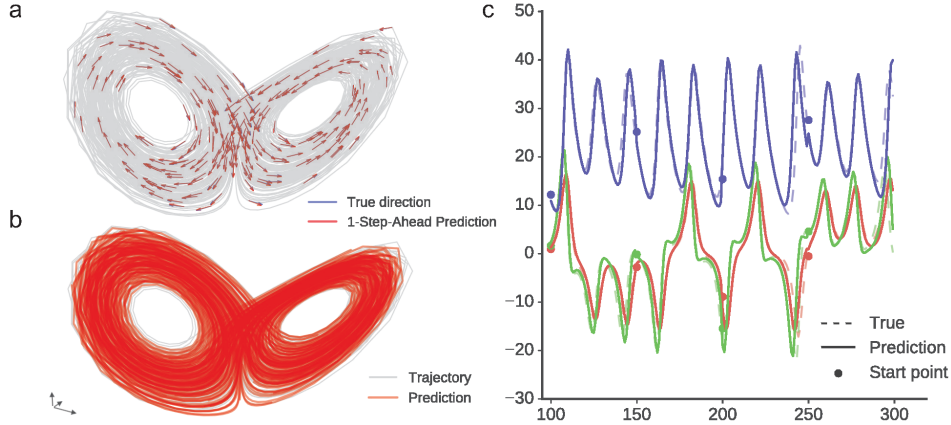

Figure 4: **(a)** Vector plot of 1-step-ahead prediction on one Lorenz trajectory (test). **(b)** 50-step prediction every 50 steps on one Lorenz trajectory (test). **(c)** A 200-step window of **(b)** (100-300). The dashed lines are the true trajectory, the solid lines are the prediction and the circles are the start points of prediction.

### 4.4 Chaotic dynamics

Chaotic dynamics (or near chaos) has been postulated to support asynchronous states in the cortex [1], and neural computation over time by generating rich temporal patterns [2, 25]. We consider the 3D Lorenz attractor as an example chaotic system. We simulate 20 trajectories from,

$$\begin{aligned}
\dot{x} &= 10(y - x), \\
\dot{y} &= x(28 - z) - y, \\
\dot{z} &= xy - \frac{8}{3}z.
\end{aligned} \tag{13}$$

The initial state of each trajectory is standard normal. The duration is 200 and the time step is 0.04. The first 300 transient states of each trajectory are discarded. We use 19 trajectories for training and the last one for testing. We train a model with 10 basis functions. Figure 4a shows the direction of prediction. The vectors represented by the arrows start from current states and point at the next future state. The predicted vectors (red) overlap the true vectors (blue) implying the one-step-ahead predictions are close to the true values in both speed and direction. Panel (b) gives an overview that the prediction resembles the true trajectory. Panel (c) shows that the prediction is close to the true value up to 200 steps.

## 5 Learning V1 neural dynamics

To test the model on data obtained from cortex, we use a set of trajectories obtained from the variational Gaussian latent process (vLGP) model [26]. The latent trajectory model infers a 5-dimensional trajectory that describes a large scale V1 population recording (see [26] for details). The recording was from an anesthetized monkey where 72 different equally spaced directional drifting gratings were presented for 50 trials each. We used 63 well tuned neurons out of 148 simultaneously recorded single units. Each trial lasts for 2.56 s and the stimulus was presented only during the first half.

We train our model with 50 basis functions on the trial-averaged trajectories for 71 directions, and use 1 direction for testing. The input was 3 dimensional: two boxcars indicating the stimulus direction $(\sin\theta, \cos\theta)$, and one corresponding to a low-pass filtered stimulus onset indicator. Figure 5 shows the prediction of the best linear dynamical system (LDS) for the 71 directions, and the nonlinear prediction from our model. LDS is given as $x_{t+1} = Ax_t + Bu_t + x_t$ with parameters A and B found by least squares. Although the LDS is widely used for smoothing the latent trajectories, it clearly is not a good predictor for the nonlinear trajectory of V1 (Fig. 5a). In comparison, our model does a better job at capturing the oscillations much better, however, it fails to capture the fine details of the oscillation and the stimulus-off period dynamics.

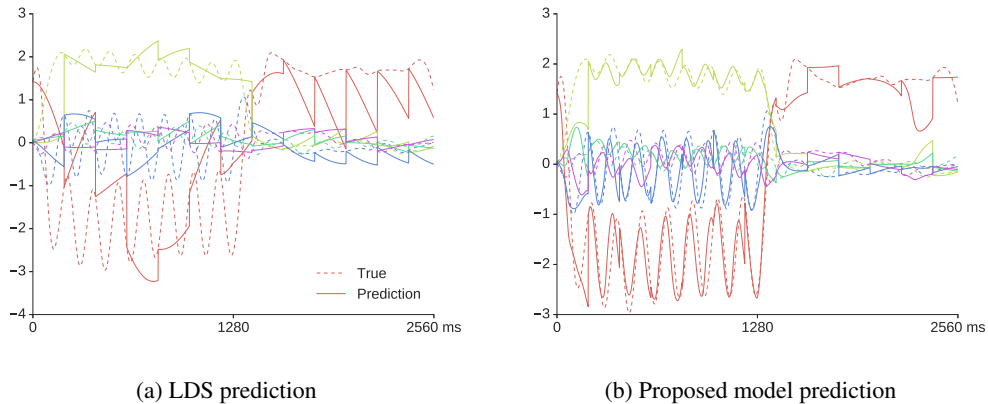

(a) LDS prediction
(b) Proposed model prediction

Figure 5: V1 latent dynamics prediction. Models trained on 71 average trajectories for each directional motion are tested on the 1 unseen direction. We divide the average trajectory at $0°$ into 200 ms segments and predict each whole segment from the starting point of the segment. Note the poor predictive performance of linear dynamical system (LDS) model.

# 6   Discussion

To connect dynamical theories of neural computation with neural time series data, we need to be able to fit an expressive model to the data that robustly predicts well. The model then needs to be interpretable such that signatures of neural computation from the theories can be identified by its qualitative features. We show that our method successfully learns low-dimensional dynamics in contrast to fitting a high-dimensional recurrent neural network models in previous approaches [17, 4, 25]. We demonstrated that our proposed model works well for well known dynamical models of neural computation with various features: chaotic attractor, fixed point dynamics, bifurcation, line/ring attractor, and a nonlinear oscillator. In addition, we also showed that it can model nonlinear latent trajectories extracted from high-dimensional neural time series.

Critically, we assumed that the dynamics consists of a continuous and slow flow. This allowed us to parameterize the velocity field directly, reducing the complexity of the nonlinear function approximation, and making it easy to identify the fixed/slow points. An additional structural assumption was the existence of a global contractional dynamics. This regularizes and encourages the dynamics to occupy a finite phase volume around the origin.

Previous strategies of visualizing arbitrary trajectories from a nonlinear system such as recurrence plots were often difficult to understand. We visualized the dynamics using the velocity field decomposed into speed and direction, and overlaid fixed/slow points found numerically as local minima of the speed. This is obviously more difficult for higher-dimensional dynamics, and dimensionality reduction and visualization that preserves essential dynamic features are left for future directions.

The current method is a two-step procedure for analyzing neural dynamics: first infer the latent trajectories, and then infer the dynamic laws. This is clearly not an inefficient inference, and the next step would be to combine vLGP observation model and inference algorithm with the interpretable dynamic model and develop a unified inference system.

In summary, we present a novel complementary approach to studying the neural dynamics of neural computation. Applications of the proposed method are not limited to neuroscience, but should be useful for studying other slow low-dimensional nonlinear dynamical systems from observations [27].

# Acknowledgment

We thank the reviewers for their constructive feedback. This work was partially supported by the Thomas Hartman Foundation for Parkinson's Research.

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
