[Supplementary Material]

## Supplement

### Wong and Wang's dynamics

$$\frac{ds_i}{dt} = -\frac{s_i}{\tau_s} + (1 - s_i)\gamma H_i \tag{14}$$

$$H_i = \frac{ax_i - b}{1 - \exp[-d(ax_i - b)]} \tag{15}$$

$$x_1 = J_{N,11}s_1 - J_{N,12}s_2 + I_0 + I_1 \tag{16}$$

$$x_2 = J_{N,22}s_2 - J_{N,21}s_1 + I_0 + I_2 \tag{17}$$

$$I_i = J_{A,ext}\mu_0 \left(1 \pm \frac{c}{100\%}\right) \tag{18}$$

where $i = 1, 2$, $a = 270(\mathrm{VnC})^{-1}$, $b = 108\mathrm{Hz}$, $d = 0.154\mathrm{s}$, $\gamma = 0.641$, $\tau_s = 100\mathrm{ms}$, $J_{N,11} = J_{N,22} = 0.2609\mathrm{nA}$, $J_{N,12} = J_{N,21} = 0.0497\mathrm{nA}$, $J_{A,ext} = 0.00052\mathrm{nA} \cdot \mathrm{Hz}^{-1}$, $\mu_0 = 30\mathrm{Hz}$.

$$\frac{d\mathbf{g}(\mathbf{x})}{d\mathbf{x}} = \mathbf{W}\frac{d\boldsymbol{\phi}(\mathbf{x})}{d\mathbf{x}}$$

$$\frac{d\mathbf{B}(\mathbf{x})\mathbf{u}}{d\mathbf{x}} = \begin{bmatrix} \mathbf{u}^\top \mathbf{W}_{B1}\frac{d\boldsymbol{\phi}(\mathbf{x})}{d\mathbf{x}} \\ \vdots \\ \mathbf{u}^\top \mathbf{W}_{Bd}\frac{d\boldsymbol{\phi}(\mathbf{x})}{d\mathbf{x}} \end{bmatrix} \tag{19}$$

Figure 6: Failure mode of unregularized locally linear model: 1 s simulation from $\mathbf{x}_{t+1} = \mathbf{A}(\mathbf{x}_t)\mathbf{x}_t + \mathbf{B}(\mathbf{x}_t)\mathbf{u}_t + \mathbf{x}_t$ model.

**Ring attractor**

Figure 7: 150 training trajectories for the ring attractor. Green circles are initial states and red circles are final states.