[Reviews · NeurIPS 2016]

Reviewer 1

Summary

The authors present a novel subclass of nonlinear autoregressive time­series model, consisting of an assumption (that the nonlinear dynamics are slow enough to be approximated using a locally linear model) and a “global leak” term (such that the state approaches the origin with an estimated timescale). The purpose of this leak term is to regularize fits to small training sets by incorporating an assumption that trajectories within the true system will occupy a finite region of the phase­space. This model is then estimated for 5 low­dimensional nonlinear dynamical systems with relevance in neuroscience and tested on held­out data, and compared (for 2 of the 5 systems) to models fit without the global leak parameterization. In general this approach seems to outperform models without the global leak term, and overall does a reasonable job of estimating fixed and ghost points of these systems. However, in several cases predicted trajectories generated by this new model seem to have qualitatively poor performance.

Qualitative Assessment

Overall the paper is technically sound, but missing some content necessary to support all the authors’ claims. The paper is most in need of more rigorous comparisons (both to ground truth, and to other state­of­the­art estimation techniques), as well as more context for interpreting the significance of the results. For instance, when examining the 1­200 step prediction of the Lorenz system, it would be useful to show whether the divergence of the estimated model’s trajectories is less than or greater than the divergence of the true system in responses to small perturbations of the initial conditions over the 200 step window. In all cases, the paper could have benefitted substantially from a more thorough comparison to other approaches for dynamical state estimation beyond just the strawman of a linear dynamical system. ​It’s unclear from reading this paper how novel the “global leak” parameterization is, and how it differs or compares to other techniques for regularized estimation of dynamical systems. Ultimately I think this parameterization might prove to be a useful advance, but at the moment it’s unclear the magnitude of that advantage. The paper was generally clearly written. The inconsistency of whether this approach was being compared to both the true trajectories and those generated by a locally linear model without the leak term, or just the true trajectories made the flow of the paper a little less cohesive.

Confidence in this Review

2-Confident (read it all; understood it all reasonably well)


Reviewer 2

Summary

This is an interesting study on nonlinear time series fitting with the aim to reconstruct the underlying dynamics and to identify fixed points and ghost points which have more interpretable meaning in neural computations. The authors fit time series from sampled nonlinear trajectories using a model with separable, linear input interaction, and radial basis functions as a nonlinear function approximator. They assume slow continuous dynamics, allowing for locally linear approximations and a model that learns the velocity field. For regularization, the model includes a global leak that pulls the model towards the origin and helps prevents blow-up. The authors demonstrate the performance of their method in a variety of applications as well as the shortcomings of their current model. As the authors argue, their method appears to provide more intuition about the underlying dynamical systems than other methods for dynamical systems analysis (e.g. fitting neural networks, recurrence plots, etc.).

Qualitative Assessment

The writing is clear and the methods are straightforward. The results look promising, though it would be extremely interesting to see the differences with other nonlinear fitting models. Indeed, the main concern I have is that the authors do not adequately demonstrate the reasons their model succeeds better than other models/methods. There is some discussion of this but not many results. Instead, the authors only compare the performance of their model only with that of linear dynamical systems and to a “locally linear model” without regularization. How does it compare to nonlinear models, and what sets this model apart from other existing nonlinear models, that sometimes “produce wild extrapolations.” Is it mainly the regularization? I commend the authors for their candid description of how their approach works and fails at the ring attractor, which is an interesting finding. Minor: Ghost points are not defined.

Confidence in this Review

3-Expert (read the paper in detail, know the area, quite certain of my opinion)


Reviewer 3

Summary

The authors propose a method of fitting nonlinear dynamical systems from trajectories. They use a parametrization with radial basis functions, and linear interactions between them. They demonstrate the method on several example tasks, showing a match of both trajectories and overall phase space flow.

Qualitative Assessment

This is an important topic, and the authors present a novel approach to it. The manuscript is clearly written, and the examples are well motivated. The detection of fixed points in test data is particularly impressive. One weakness is that all examples are very low dimensional, and it is not clear whether and how the method will generalize to this regime.

Confidence in this Review

3-Expert (read the paper in detail, know the area, quite certain of my opinion)


Reviewer 4

Summary

They propose a class of dynamical systems models for neural data, such as firing rates, single neuron biophysics, etc. This allows the visualization of a velocity field, where one can see features such as fixed points and basins of attraction. These are tested on simulations of models and compared to the ground truth.

Qualitative Assessment

This looked technically correct and well written. Some analysis of how well it works with different sampling rates might be useful.

Confidence in this Review

1-Less confident (might not have understood significant parts)


Reviewer 5

Summary

Fitting models of dynamical systems from data has a long tradition in Nonlinear Dynamics. If successful, it holds promise to provide important insights into neuronal systems. The authors propose an ansatz, which goes only slightly beyond the many previous approaches using radial basis functions in that it includes a decay to origin, which appears to represent a useful regularization.

Qualitative Assessment

The authors seem to mostly aim at phase portraits in 2 dimensions, which I consider of restricted utility. The method is shown to work for some 2-dimensional examples but gives partially wrong results for the ring model. An application is shown, where it the method predicts experimental data better than linear models, which is not really a surprise, and which I suspect to depend on the particular choice of parameters. Unfortunately, this most interesting last part of the paper lacks the details necessary to judge it.

Confidence in this Review

3-Expert (read the paper in detail, know the area, quite certain of my opinion)


Reviewer 6

Summary

The authors demonstrate a fitting approach using basis functions and a regularization term to robustly fit nonlinear dynamical systems. The inclusion of the global regularization term is their main contribution, and they clearly illustrate how it makes their procedure robust against test cases outside of the training range (Table 1 & Fig 6 in supplementary material). They demonstrate their approach for various types of dynamics: a fixed-point attractor, FitzHugh-Nagumo, a ring attractor and Lorentz chaotic dynamics. They also very briefly discusses how the approach can be applied to neural recordings.

Qualitative Assessment

Overall I found the paper to be solid and rather enjoyable, and I would qualify it as a strong candidate for a poster. The authors' method of plotting velocity fields by decomposing the velocity into direction and speed, which they've apparently introduced, is especially effective. It made their arguments and conclusions much easier to follow, and will hopefully be picked up by others. In my opinion stating that this approach leads to “interpretable models” might be somewhat overselling the results – the interpretability of the results is still hampered by the fact that models are composed by 10-100 more or less arbitrary basis functions. That being said, their capacity to reproduce salient features of the phase diagram certainly makes them more interpretable than, say, recurrent neural networks. *** Update following authors' response *** "Interpretable model" can mean many things. For instance, it can mean that the value of each c_i, σ_i,… is interpretable – the meaning I originally had in mind – or that the model results in an interpretable phase diagram – the meaning the authors had in mind. It should suffice to define the desired meaning the first time this expression is used. Discussions are for the most part appropriately detailed, although sometimes they feel somewhat condensed, perhaps due to the large number of examples the authors discuss within their allowed 8 pages. Specific issues are noted below. *** Section 2 — Model - The linearization in eq. (2) seems incorrect to me. One typically linearizes an update function (F here) around some other point (say x*) in the neighborhood of x_t, so that we have something of the form x_{t+1} = x_t + A(x*)(x_t - x*) + B(x*)(x_t - x*)u_t + ε_t We can then collect the constant terms as b(x_t) = -A(x*)x* - B(x*)x* Note that it is critical that x* and x_t be different in general, otherwise the linear terms are always zero. This distinction seems to be made in the given ref. [9], but it is not done here. Thus eq. (2) needs to be corrected, or if it is correct, a more detailed derivation needs to be given. These issues are further illustrated by the comment that A is not unique: if A(x_t) is already defined as the Jacobian of F_x at x_t, how can it not be unique ? Fixing the derivation of (2) should help clarify this comment. *** Update following authors' response *** I think the issue boils down to clarifying that b(x_t) and B(x_t) are not linearizations around x_t, but rather around some other point h(x_t). It would then be clear that they are not unique, as they depend on the function h. One should probably write b(x)=Fx(h(x)) on line 38. - line 39: B should be a function from ℝ^d → ℝ^d x ℝ^{di}. - The previous issue might also be addressed by simply removing the first part of section 2 up to “redundant and over parametrizes the dynamics.” Indeed, this whole section is never used for the rest of the paper. I found eq. (3) a clearer starting point for the arguments than eq. (2). - The authors should state that W_g, W_B and τ all depend on the step size Δ. - Eq. (4). I feel that the regularization term should be exp(-τ² x_t). If it is correct as is, the authors should comment on the benefit of writing the regularization constant in this way (rather than just using a constant β ≡ exp(-τ²) ). - line 61: Unless B is constant, the Jacobian should also include a term ∂B/∂x. - line 63: As far as I am aware, “ghost point” is not a frequent term in studies of dynamical systems. Perhaps the authors could provide a reference, or a short definition ? Same comment regarding “slow point”, which first appears on line 92. *** Section 4.2 — Nonlinear oscillator model - Since the text refers to figures 2a-2d, the subfigures should show alphabetic labels. *** Section 4.3 — Ring attractor dynamics To me this is the most interesting of the presented examples. The fact that only a subset of qualitative features are reproduced makes this a good test case to explore what this fitting procedure can and cannot do. Unfortunately the authors limit their discussion to a description, simply stating that the fit and true models show qualitatively different behavior. I think it would be worth trying to distill a what parts of the dynamics are difficult to fit and why. For instance, why is it that the radial dynamics seem to completely dominate Fig 3a, even along the ring (where dr/dt should be almost zero) ? - Fig 3b : As far as I can tell, there are in fact two fit and two true trajectories in this figure (distinguished by dashed and full lines). The legend says there is one of each. *** Section 4.4 — Chaotic system - Fig 4c : The true trajectories are almost impossible to see. Also the line styles in the legend should match those in the figure. *** Section 5 — Learning V1 neural dynamics This section feels incomplete. The LDS model used is claimed to be the “best”, but no justification or reference is given to support this claim. In fact it is not even stated which LDS model is used – this could be included in the supplementary material, similar to what was done for section 4.1. The example studied in this section is by far the paper's most complex, and yet its explanation is one of the shortest. Consequently, it is difficult to picture what it is exactly that the model is fitting. Ideally a figure would be added to illustrate the inputs, but at the very least the second paragraph should be lengthened to include a more precise description of the inputs. As it stands, it is not even specified what time of recording the data stems from (only that it is “large scale”). *** Supplementary material - As far as I understand, the interest in giving eq. (15) in terms of the x_i is to relate these dynamics to physical quantities. Since the authors never do this in the text, I would remove the middle expression in eq. (15). Similarly, I think defining x_2 in eq. (16) is superfluous; moreover, x_1 is never properly defined. The expressions in terms of s_1 and s_2 already appear as-is in ref. [20]. Also, the parameters J_{A,ext} and c' which appear in eq. (17) are never defined. - In fig 7, green and red dots should be above the blue lines rather than below. *** Grammar Some minor grammatical typos were noted, which in no way affect the quality of the paper. Possible corrections are listed here for the author's reference. 4: Our model incorporates a prior assumption 7: bifurcations 17: dynamical systems [5,6] 27-28: These features encourage the model 36: of an autoregressive 40: in the multivariate RBF 60: dynamics are locally 61: via the Hartman-Grobman 82-83: where the tasks typically involve 150: The initial state of each trajectory is picked from a standard normal 151: We use 19 trajectories 152: and the last one for testing. 166: The input was 3 dimensional 173: This end of sentence could be better formulated: “robustly predicts well.”

Confidence in this Review

2-Confident (read it all; understood it all reasonably well)